# The Relative Merits of an Individualized Versus a Generic Approach to Rating Functional Performance in Childhood Dystonia

**DOI:** 10.3390/children8010007

**Published:** 2020-12-25

**Authors:** Hortensia Gimeno, Jessica Farber, Jessica Thornton, Helene Polatajko

**Affiliations:** 1Complex Motor Disorders Service, Paediatric Neurosciences, Evelina London Children’s Hospital, Guy’s & St Thomas’ NHS Foundation Trust, London SE1 7EH, UK; 2Department of Psychology, Institute of Psychiatry, Psychology and Neurosciences, King’s College London, London SE5 8AF, UK; 3Department of Occupational Science and Occupational Therapy, University of Toronto, Toronto, ON M5G 1V7, Canada; jessica.farber@utoronto.ca (J.F.); jessica.thornton@utoronto.ca (J.T.); h.polatajko@utoronto.ca (H.P.)

**Keywords:** hyperkinetic movement disorders, dystonia, performance, deep brain stimulation, CO-OP Approach, performance quality rating scale

## Abstract

Aims. The Performance Quality Rating Scale (PQRS) is an observational measure that captures performance at the level of activity and participation. Developed for use with the Cognitive Orientation to daily Occupational Performance (CO-OP), it is a highly individualized approach to measurement. CO-OP is currently being studied in childhood-onset hyperkinetic movement disorders (HMD) and deep brain stimulation. The purpose of this study was to compare two different approaches to rating performance, generic (PQRS-G) and individualized (PQRS-I), for children with childhood-onset hyperkinetic movement disorders (HMD) including dystonia. Method. Videotaped activity performances, pre and post intervention were independently scored by two blind raters using PQRS-G PQRS-I. Results were examined to determine if the measures identified differences in e performance on goals chosen by the participants and on change scores after intervention. Dependent t-tests were used to compare performance and change scores. Results. The two approaches to rating both have moderate correlations (all data: 0.764; baseline: 0.677; post-intervention: 0.725) and yielded some different results in capturing performance. There was a significant difference in scores at pre-intervention between the two approaches to rating, even though post-intervention score mean difference was not significantly different. The PQRS-I had a wider score range, capturing wider performance differences, and greater change between baseline and post-intervention performances for children and young people with dystonic movement. Conclusions. Best practice in rehabilitation requires the use of outcome measures that optimally captures performance and performance change for children and young people with dystonic movement. When working with clients with severe motor-performance deficits, PQRS-I appears to be the better approach to capturing performance and performance changes.

## 1. Introduction

Rehabilitation approaches for childhood-onset hyperkinetic movement disorders (HMD) continue to fail to be systematically evaluated. This is, in large part, due to the lack of appropriate measures to objectively capture performance and changes in performance in activity and participation that are important to the children, young people and their families [1]. Measuring change of pharmacological, surgical, and rehabilitation interventions in childhood-onset HMD including dystonia is also challenging given the heterogeneity of this population [2] and the lack of validated assessments currently available for use in research and clinical practice [3].

Childhood-onset HMD are neurological disorders that have unwanted or excess movements known as hyperkinetic movements [4]. Dystonia is the most common type of HMD and is characterized by sustained muscle contractions and resulting in repetitive and twisting muscle movements [5]. Considering dyskinetic cerebral palsy (CP) is the largest group [4] in this heterogenous group, evidence on pharmacological and neurosurgical management approaches is largely not evidence based and provides only with limited improvements in the reduction of dystonia [6]. Rehabilitation interventions for this population are lacking and are urgently needed. The most commonly used measures to capture change with these interventions are tools designed to quantify impairment, not performance at the level of activity and participation [7], further, these tools are of questionable value as outcome tools, they are not sufficiently sensitive to capture performance change on activities that are important for children, young people, and families [8].

Cognizant of the need to objectively quantify performance and capture change, a research group developing the Cognitive Orientation to daily Occupational Performance (CO-OP) Approach created the Performance Quality Rating Scale (PQRS), an observational performance rating tool. CO-OP is an individualised task-oriented approach that uses cognitive strategies to improve motor performance in important, self-selected daily activities. In CO-OP the use of a global cognitive strategy (Goal-Plan-Do-Check) is introduced for the young person to use it throughout to solve difficulties encountered when performing any of the self-selected goals. The therapist uses guided discovery enabling the patient to come up with their own solutions to their difficulties. The PQRS allows for the objective evaluation of performance in these activities.

Consistent with observational approaches to evaluating performance, the PQRS use a ten-point rating scale, to objectively quantify performance; a scoring guideline is created to individually tailor the rating scale to fit the specific performance goal of the individual receiving intervention. Raters, in vivo, or off video recordings, rate the observed performance using the scoring guideline [9].

The PQRS has been used in a number of research studies evaluating the CO-OP Approach [10], with several child and adult populations including developmental coordination disorder (DCD), spastic cerebral palsy (CP), stroke, and acquired brain injury [10]. In all cases, the PQRS was able to capture changes in performance over time. Further, in the two-group design studies, the PQRS was able to capture significant group differences in outcomes; differences that were support by other measures in a number of the studies.

The extensive use of the PQRS in intervention studies notwithstanding, there have been few standalone investigations of the measure. Psychometric information on the PQRS, such as inter-rater reliability, has generally been reported as part of the intervention studies. This shortcoming was noted in a 2012 review of upper extremity assessments for stroke and CP, where the PQRS was excluded because of the lack of published psychometric properties [11]. This was partially corrected in 2015, when Martini and colleagues published an examination of two PQRS systems. They reported that the PQRS showed substantial reliability and internal responsiveness in two different populations, adults with stroke and children with DCD [12].

One of the systems described by Martini et al. (2014) is a generic form of the PQRS (PQRS-G), where performance was rated on two dimensions: degree of task completion and the quality of the performance [12]. The former is scored relative performance criteria for the task while the latter is scored relative some presumed normative, or typical performance of the task. These scores are then combined to yield a single PQRS-G score. For example, an individual with dystonia wanting to learn to carry a glass of water across a room without spilling would need to carry the glass across the room without spilling a drop, holding it in a ‘typical’ way and carry it to earn a (10) top score. If the individual accomplishes the task by carrying the glass in an ‘atypical’ manner, perhaps by pressing the glass against the chest while walking, the score would not be 10, even though not a drop was spilled.

While there may be some rationale for an approach to rating performance relative some sense of normal or typical performance, this approach, the PQRS-G, is at variance with the approach to scoring that had been used originally [13] and in the majority of the studies of the CO-OP Approach [10]. In most studies, an individualized criterion-based approach had been used where the participant determined the criterion for successful activity performance. In this individualized approach, referred to here as PQRS-I, the individual with dystonia described above would rate a score of 10 if he carried a glass of water across the room without spilling a drop, even if he pressed the glass against his chest while walking. This difference was observed during a recent proof of concept study using the CO-OP Approach in childhood-onset HMD, after Deep Brain Stimulation (DBS) [14,15,16]. It should be noted that when the PQRS was first created it was done for a DCD population whose performance issues are quite different from those with dystonia. For a child with DCD the PQRS-G and PQRS-I scores would likely be the same. This distinction has only become apparent one once the CO-OP Approach, and in tandem the PQRS, started to be used with populations with more sever motor impairments such as individuals with stroke or cerebral palsy.

The inherent value of a PQRS-G is that it rates performance relative some imagined norm, the inherent value of the PQRS-I is that it rates performance relative the individual’s criterion for goal achievement. Both have merit, depending on context. Of interest here is how the two approaches compare when used with the HMD population where it may be expected that performance will be successful, i.e., meets the criteria expectations of the individual, but will not meet normative or typical performance expectations.

The purpose of this study was to examine the differences between the two PQRS scoring systems in capturing performance and performance changes for childhood-onset HMD including dystonia. This was done by comparing the PQRS-G scores to the PQRS-I scores.

## 2. Method

### 2.1. Research Design

A secondary analysis of data from a larger study evaluating the CO-OP Approach™ with young people with HMD and DBS [14,16] was undertaken.

Two blinded raters (J.T. and J.B.) watched and rated 170 videos of nine young people with childhood-onset HMD performing self-selected goals presented randomly at two time points, baseline and post-intervention. Goals ranged from self-care activities such as toothbrushing, applying eye mascara, making a drink, buttering toasts, to leisure activities such as swimming or riding a bike.

Ethics approval for the present study was obtained by the Research Ethics Boards of the participating institutions. The primary study was approved by the NHS Health Research Authority Oxford A Research Ethics Committee (14/SC/1159) and was registered with the ISRCTN Registry (ISRCTN57997252).

### 2.2. Data Source

The videos came from the recordings of nine young people with childhood-onset HMD enrolled in a CO-OP intervention study. Participants had been prospectively recruited using the Complex Motor Disorders Service (CMDS) database at Evelina London Children’s Hospital in London, UK, as part of the primary study investigating the CO-OP Approach™. Participants ranged in ages from 9–18 with confirmed diagnoses of HMD with DBS implantation [14,16] and demographics are presented in Table 1.

Videos for scoring were randomly selected by a research assistant outside the core research team, from the data set of nine children whose main data have been reported elsewhere [16]. These videos, which comprised of ten randomly selected videos per child (five goals each rated at two time periods, baseline, and post-intervention), were randomly presented to the rating authors. This resulted in a sample of 85 videos assessed using both PQRS-I and PQRS-G (170 available scores), each system applied by a different rater. Raters viewed all videos in a randomized order in several rating sessions (i.e., the same child videos were randomly presented and therefore pre- and post- videos not necessarily rated on the same rating session).

Inclusion and exclusion criteria. To have been included in the parent CO-OP Approach™ study, and subsequently be captured on the videotaped performances used here, individuals must have (a) been willing to participate; (b) been able speak and understand English; (c) been able to follow simple instructions and to engage in intervention; (d) had a diagnosis of HMD with DBS implantation; (e) been 6-21 years of age; (f) had developing skills in self-care; (g) been able to mobilize independently; (h) required adult assistance to complete activities appropriate for age (such as dressing, eating and drinking, making a bed); (i) had cognitive ability of age 6; and (j) had IQ score of above 70 as assessed by the Wechsler Intelligence Scale for Children (WISC).

Participants were excluded based on the following criteria: (a) having a condition that presents with pure spasticity; (b) having dystonia that is a result of a neurodegenerative condition; and (c) having any surgery scheduled during the study period.

### 2.3. Measures

PQRS-G: The PQRS-G (generic, as described by Martini and colleagues (2015)) is an observation tool used to rate the degree of task completion (where 1 is ‘0% task completion’, and 10 is ‘100% task completion’) and the quality of the performance (where 1 is ‘n/a’ as cannot complete any part of the task and 10 is ‘excellent’). The individual and therapist collaborate to create an operational definition of what a ten would look like in terms of the task completion. Quality of performance is rated relative to some presumed normative or typical performance of the task. These two scores are then averaged to compute the final score (for further information please see the original publication by Martini and colleagues [12]).

PQRS-I: The PQRS-I (individualized) is an observation tool used to rate task performance relative the desired outcome where 1 is ‘can’t do it at all’ and 10 is ‘can do it very well’, see Table 2. The individual and therapist collaborate to create operational definitions of what a ten would look like in terms of the desired performance and the performance is relative to that definition.

Procedure. Two authors (J.F. and J.T.) rated video data. Prior to initiating video rating, the raters received training from the senior authors (H.P. and H.G.) and established inter-rater reliability with an expert rater for absolute agreement using intra-class correlation (ICC) with both PQRS-I (ICC = 0.830) and PQRS-G (ICC = 0.883). Both raters scored the same videos, each using a different PQRS rating scale. Therefore, each rater scored 5 randomly presented goals per participant (*n* = 9 participants) at baseline and post-intervention. Each rater therefore rated 45 goals at baseline and 40 at post-intervention (missing data for one case at post-intervention).

### 2.4. Data Analysis

Data analysis was completed using IBM statistics for MAC, version 26.00. Descriptive statistics of mean and standard deviation (SD) were used to summarise scores at baseline and post-intervention using the PQRS-I and PQRS-G scales.

For comparison of the two scales, mean scores were calculated for baseline and post-intervention scores and dependent t-test was used to determine whether the means of two groups were statistically different from one another at both time points (baseline and post-intervention).

Correlations between both systems (all data) were first completed and a scatterplot evaluated for differences in the systems at either time point. Separate analysis of correlations between both systems at baseline and post-intervention was then completed.

## 3. Results

A total of 170 scores was available for PQRS-I and PQRS-G (85 goal scores for each PQRS system). Baseline scores for the PQRS-I (mean 3.51, SD: 2.36) were lower than those for the PQRS-G (mean 4.11, SD: 1.74) and statistically significant (*p* = 0.002). Post-intervention scores, on the other hand, were higher for the PQRS-I (mean 6.16, SD 2.73) than the PQRS-G (mean 5.89, SD 1.72) but not statistically significant (*p* = 0.181) as shown in Figure 1.

These results may indicate that performance could be captured differently by the two observational tools. When comparing all PQRS-G data to PQRS-I data, the ICC is 0.830, 95% Confidence Intervals (CI) (0.74–0.89), which indicates a “moderate” agreement. In the analysis of performance, results indicated that at the baseline time period, PQRS-I and PQRS-G ICC is 0.77, 95% CI (0.63–0.85). At post intervention, ICC is 0.79, 95% CI (0.68–0.86).

At post-intervention time period, however, the mean scores, 5.88 for PQRS-G and 6.2 for PQRS-I, are not significantly different from one another, likely due to high variance between the scores. Figure 2 shows that PQRS-G is higher at the low values, and lower at the high values, therefore more constrained in its distribution. PQRS-G never starts at 1 and never goes up past 8 or 9. PQRS-I has more spread from 1 to 10. Regression analysis between the two PQRS scoring systems showed R^2^ = 0.583 for all data available indicating moderate agreement and shown in Figure 2. When regression analysis was done per phase, baseline R^2^ was 0.458 for baseline phase and R^2^ = 0.525 for scores post-intervention. This indicates that, at baseline, the extend of the variance of one tool does not explain the variance of the other tool. For post-intervention this variance is moderate.

Pearson’s correlations were significant (*p* < 0.001) for all data (0.764), baseline (0.677), and post data (0.725).

In the analysis of performance change, both tools showed a significant change from baseline to post-intervention time periods. However, change scores for PQRS-I was 2.6 (95% CI 1.75–3.43) as compared to 1.7 (95% CI 1.31–2.15) for PQRS-G (both significant change, *p* < 0.001). These findings indicate that PQRS-I demonstrated a significantly greater positive change in scores from baseline to post-intervention than did PQRS-G (mean 0.863, SD 2.94, 95% CI 0.24–1.49; *p* = 0.007).

## 4. Discussion

The PQRS is a potential individualized tool for the evaluation of performance and performance change, and has been used to objectively evaluate performance in studies of the CO-OP Approach™ in a number of populations [9,17,18,19,20]. Two approaches to rating performance are reported here, one informed by notions of typical performance (PQRS-G) [12] the other restricted to client performance criteria (PQRS-I). Both approaches have merit—depending on context, i.e., the nature of the performance of interest. This study sought to examine the differences these two approaches yield for individuals with childhood-onset HMD including dystonia.

In reviewing the findings of this analysis, the PQRS-I may be the better approach to rating for this population. The PQRS-I captures performance and performance change in an individualized way, without penalizing the individual for successful, yet non-typical performance, as can be seen in those with HMD, or dystonic movements. Using PQRS-I, an individual would get a top score if they had reached their personally defined goal criteria, even though the task may be performed in an atypical/adapted manner. Whereas in using the rating on the PQRS-G, which scores performance based on a usual or typical way of performing a task, the individual who accomplishes the desired goal in an atypical manner would never achieve a 10. In other words, the PQRS-I allows for individual differences in performance, whereas PQRS-G does not. In addition, this study provides support for the PQRS-I as a potential scoring system to use alongside CO-OP Approach™ in other populations where the aim of an intervention is not to achieve normal or typical task performance.

The PQRS-G and PQRS-I score performance in very different ways. As a result, they capture performance and performance change differently. Findings indicate that PQRS-G and PQRS-I differ in the ability of the individual to achieve a top score. In this population, a top score of 10/10 was impossible to achieve using the PQRS-G due to judgment of performance being based on normative ways of performing the task. PQRS-I allows for top scores of 10/10 despite dystonic movement and/or postures.

Findings also suggest that PQRS-G is less likely to show very high scores due to the averaging that is involved in the scoring process. As reviewed previously, PQRS-G involves averaging the scores given out of 10 for both completion of the task, and quality of the performance. So, if they achieve a 10/10 score for completion, and only a 7 or 8 out of 10 for quality, the averaging of these two PQRS-G scoring sections makes it impossible to achieve a top score.

In contrast, the PQRS-I does not involve averaging across two scoring sections, allowing for a greater access to the whole range of possible scores. This results in lower overall scores at baseline, and top scores achievable. Thus, PQRS-I has a wider range of possible scores.

The PQRS-G and PQRS-I perform differently in certain circumstances. For example, when an individual completes the task with poor quality, scores will be higher using PQRS-G than with PQRS-I. Considering the often-inflated scores at baseline due to averaging of completeness and quality scores, and inability to achieve top scores PQRS-G shows less change from baseline to post-intervention than PQRS-I. PQRS-I overall showed lower scores at baseline, and higher post-intervention scores, and therefore showed greater change when measuring the same performances.

Although the PQRS-G and PQRS-I have some similarities as PQRS scoring systems, they measure performance in very different ways. For this population, who will likely never be able to perform a task in a normative or usual way, the PQRS-I provides a scoring tool that will not penalize them for their disability. The unpredictability of movements in young people with HMD impacts the performance, but it does not mean that they cannot achieve goals that are meaningful and relevant to them. When an individual achieves 100% of their goal, this may not be captured by the PQRS-G because of comparisons to typical or usual performance. In contrast, findings suggest that the PQRS-I is more sensitive to meaningful functional achievements.

To use the PQRS-I in practice or for future research, it is recommended that clients and clinicians work together and take time to develop a clear operational definition of the client’s goal prior to initiating evaluation. This will help the clinician to score performance quality based on the client’s understanding of success and completion. Performance should not be penalised for the presence of dystonic movement and/or postures, or in other populations, for limitations to performance that are inherent given their specific impairment. Whilst dystonia might be present, improvement in performance can be achieved even with underlying dystonia evidently visible. Similarly, unusual performance quality related to the individual’s impairment should not be penalized (e.g., holding a bowl in a non-normative way to compensate for dystonic movement and/or postures). Anecdotally, a young person in our study put it ‘my goal is to carry a bowl of cereal, not to carry a bowl of cereal without showing any dystonia’.

Impromptu discussions with young people during the course of our study revealed their preference for the use of PQRS-I so that performance could be scored rather than receiving penalisation for their involuntary movements.

This study is not without limitations. Findings may not be generalizable beyond this specific population. Children and young people with HMD including dystonia experience a unique set of challenges and strengths, which may or may not be similar to those with other diagnoses or conditions. Furthermore, the participants included here are a heterogeneous group of disorders. Even though the sample size included here is small, in terms of participants (*n* = 9), the sample size for data points is considerably larger with *n* = 170 data points included. This is not without limitations as data points are not independent and the same child would have had up to 5 goals (and videos of such performance) before and after intervention.

## 5. Conclusions

The PQRS-I may be the optimal tool for capturing performance and performance change with this population. It appears to capture a broader range of performance and, in turn, has the potential to yield larger change scores. PQRS-I appears to allow for a wider range of performance scores and for the achievement of top scores despite condition-specific impairment. This tool appears to be better for use with clients who achieve their goal despite the dissimilarity in performance when compared to the normal or typical way of performing.

## Figures and Tables

**Figure 1 children-08-00007-f001:**
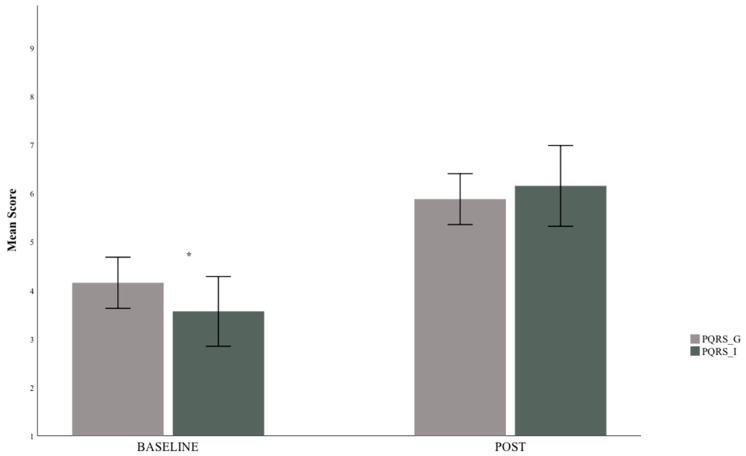
Comparing means at Baseline and at Post-Intervention. * indicates significant difference. PQRS-G = Performance Quality Rating Scale—Generic; PQRS- I: Performance Quality Rating Scale—Individualised.

**Figure 2 children-08-00007-f002:**
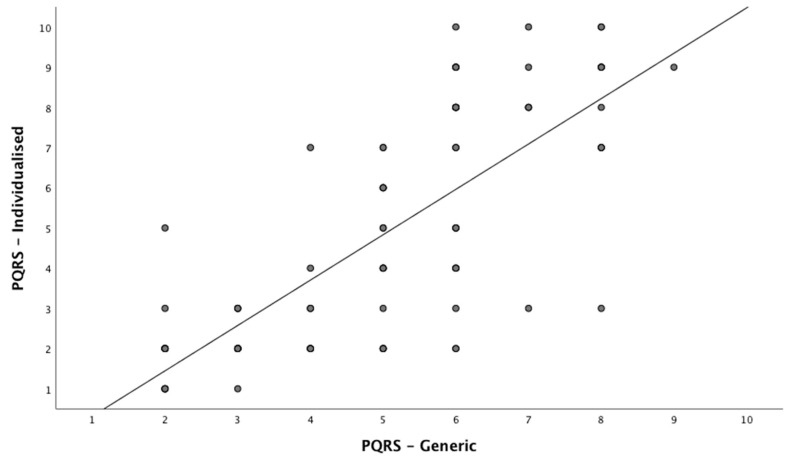
PQRS-G vs. PQRS-I scores across all time periods.

**Table 1 children-08-00007-t001:** Demographics.

Child	Age	Sex	Diagnosis	Aetiology	GMFCS	MACS
1	9 years 8 months	Male	Primary unknown	Idiopathic	I	II
2	17 years 4 months	Female	CP post meningitis	Acquired	I	II
3	17 years 6 months	Male	CP HIE	Acquired	II	III
4	15 years 2 months	Male	CP Ex-prem	Acquired	IV	IV
5	13 years 11 months	Female	CP HIE	Acquired	II	II
6	18 years 11 months	Female	BHC	Inherited	I	II
7	13 years 10 months	Female	CP Kernicterus	Acquired	II	IV
8	13 years 6 months	Male	Primary unknown	Idiopathic	I	III
9	12 years 3 months	Female	Primary unknown	Idiopathic	III	II

Abbreviations: BHC = benign hereditary chorea; CP = cerebral palsy; DBS = deep brain stimulation; Ex-prem = extreme prematurity; GMFCS = Gross Motor Function Classification System; HIE = hypoxic-ischemic encephalopathy; MACS = Manual Ability Classification System.

**Table 2 children-08-00007-t002:** PQRS-I Rating Scale.

Instructions:Watch video: Rate on a scale from 1 to 10 where 1 is not performed at all and 10 is performed to meet client’s goal. Video can be watched only twice.
Guidelines: Operational definitions are to be created for each client chosen goal prior to scoring.Scores are not to be based on ideas of normative performance, rather on performance criteria.Do not penalize for dystonic movement, i.e.; holding the bowl in a non-normative way to compensate for impairment.Consider false starts, length of time or any other factors that may impede function (struggling/effort).Only score what has been defined in the goal, not what is portrayed in the video (i.e.: if the video includes picking up and carrying shoes before initiating tying the laces).For multi-step goals, roughly divide each step and mark quality of performance for each, then use clinical reasoning overall. For example, in a three-step goal, each step is worth approximately 3.33/10 and would be scored in this way.For single-step goals, consider the scale as a percentage of completion in addition to “functional quality”.For a goal with concepts such as “not spilling” a 10 would be no spilling at all, and the score would be reduced depending on the severity of the spilling (e.g., assign an 8 for mild spilling and a 1 for severe spilling).
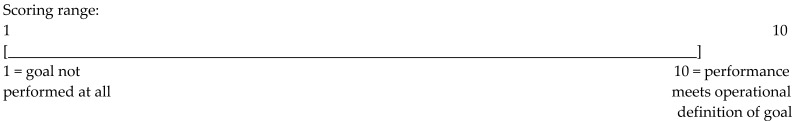

## Data Availability

Our ethics agreement prevents data being openly available, but individual researchers may request anonymized data.

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
