# Peer review of "The Relative Merits of an Individualized Versus a Generic Approach to Rating Functional Performance in Childhood Dystonia"

_children, 2020, doi:10.3390/children8010007_

Round 1

Reviewer 1 Report

The authors report on the use of two PQRS methods (generic and individualized) to assess perceived outcomes. The study identified that the individualized approach provides a larger change - a more granular evaluation.

There are multiple concerns though with the study. First, the authors cite Martini et al and effectively replicate a similar evaluation without clearly stating that. Martini et al indeed performed a similar evaluation in a different population. Although the authors allude to this point, they don't make it clear and sometimes report as if Martini et al only evaluated the generic format. Second, unfortunately I do not have access through my institution to many of the references including ones crucial for method development. As a result, I am not sure if the method have been reported previously and can be significantly reduced and simply cited. Third, the authors note that the different between the mean scores of the generic and individualized methods are not statistically significant but proceed to assert that individualized score is better using the same logic as in the introduction but this study does not provide direct evidence that an individualized score is better. The only conclusion we can make is that the score range is wider, as expected, in the individualized version than the generic version (intrinsic property of the test). So I don't agree with the authors' rather generous interpretation of the data. Furthermore, the limitation section is inadequate and needs to be developed. The conclusions and discussion need to be notably adjusted and adjusted to match the results.

The authors use the term significant inappropriately in the results section of the abstract.

The authors cited three personal citations when I think they could have limited that to one or maybe two.

Author Response

Thank you to the reviewer for the insightful comments. Below we have re-written the reviewer comment and our response. 

Reviewer: First, the authors cite Martini et al and effectively replicate a similar evaluation without clearly stating that. Martini et al indeed performed a similar evaluation in a different population. Although the authors allude to this point, they don't make it clear and sometimes report as if Martini et al only evaluated the generic format.

Author’s response: Thank you for this observation. Ours is indeed a similar evaluation to that of Martini and colleagues investigating the PQRS– i.e., investigating two different approaches to scoring the PQRS using a similar design – but we do not consider it a replication - we were not looking to replicate their findings.  Martini and colleagues were interested in the differences between PQRS– generic vs PQRS – operational definition, i.e., the difference between a generic vs a specific approach to scale definition. We were interested in the differences between PQRS– generic vs PQRS – individualized, i.e., the difference between a generic (quasi norm-based) approach vs a client-driven approach to scale definition (an approach that had been used in the early CO-OP studies.) Our empirical observations had suggested that this would be important for individuals with severe motor problems. We have added commentary about Martini and colleagues exploring two different systems to the manuscript to make this point clearer.

Reviewer: Second, unfortunately I do not have access through my institution to many of the references including ones crucial for method development. As a result, I am not sure if the method have been reported previously and can be significantly reduced and simply cited.

Authors’ response: Re reviewer unable to access references for methods (16-18). These refer to the protocol papers for the studies and the results paper for the parent study from which the data for this study were drawn. The study reported here is a secondary study, drawing data from the main study, the methods for the study being reported here have not been published anywhere else and thus cannot be significantly reduced. We could, however, reduce inclusion and exclusion criteria and refer to the published study protocol though we have concerns about doing this as others might also have difficulty in accessing these papers as this might have been the case for the reviewer here.

Reviewer: Third, the authors note that the different between the mean scores of the generic and individualized methods are not statistically significant but proceed to assert that individualized score is better using the same logic as in the introduction but this study does not provide direct evidence that an individualized score is better. The only conclusion we can make is that the score range is wider, as expected, in the individualized version than the generic version (intrinsic property of the test).

So I don't agree with the authors' rather generous interpretation of the data.

Authors’ response: Re: difference between mean scores of the -g and the -I not being statistically significant. We have now clarified this further in the text. The mean scores at baseline are significantly different but they are not significantly different at post-intervention; and therefore, we believe there is merit in exploring when the -I will be used rather than the -g system. Our findings also indicate that the response from PQRS-I was significantly greater in changes from baseline to post-intervention than the PQRS-G (p=0.007), which is outlined in the last paragraph of the results section. In reviewing the outputs, we realise that the 95% confidence intervals were incorrect and I have now amended these. 

Reviewer: Furthermore, the limitation section is inadequate and needs to be developed.

Authors’ response: Re: limitation section requiring development. We have now expanded this section.

Reviewer: The conclusions and discussion need to be notably adjusted and adjusted to match the results.

Authors’ response:  this has been done

Reviewer: The authors use the term significant inappropriately in the results section of the abstract.

Authors’ response: Re: inappropriate use of term significant in abstract. We have now made corrections to this.

Reviewer: The authors cited three personal citations when I think they could have limited that to one or maybe two.

Author’s answer: Thank you for this comment. We will be willing to eliminate redundant citations and/or swap them with more appropriate ones if we were given more direction. In the manuscript, one of the authors is the developer of the CO-OP approach and previous versions of the PQRS and therefore we feel these are appropriate. We also reference the full protocols of the parent studies and the results of the parent study. One of the authors has also made significant contribution to the field of movement disorders and the papers referenced here are some of the few available that measure beyond impairment in children and young people with dystonia. Please if we could be told which papers are cited that are not appropriate, we will be happy to amend our manuscript accordingly.

Reviewer 2 Report

The relative merits of an individualized versus a generic approach to rating functional performance in childhood dystonia

  • The paper is well written, with sound methodology, comparing PQRS-G and PQRS-I, which is an important topic.

A few comments could be made about the paper.

  • I suggest that the introduction should be made more concise and simpler to follow.
  • There is a relatively small number of patients in the study, even for a rare group of disorders.
  • The study population is heterogenous, for example BHC and CP are quite different, yet they are grouped together. The description of the limitations of the study is very brief.
  • The authors should comment whether the greater change for PQRS-I over PQRS-G meant that PQRS-I was more accurate?
  • Further clinical details may be helpful for all patients. No clinical details are given for Child 1 (Table 1).
  • It would be helpful for the authors go into more detail into what the intervention was.
  • Could the authors discuss inter and intra-rater variability?

Author Response

Reviewer: Thank you for your comments. We outline our responses to the individual comments below. 

I suggest that the introduction should be made more concise and simpler to follow.

Author’s response: thank you. We have made some changes to the introduction.

Reviewer: There is a relatively small number of patients in the study, even for a rare group of disorders.

Author’s response: thank you. The focus of this study in on the difference in scores using a “G” rating vs an “I “rating – thus the number of interest really is the number of skills these ratings were applied to. Although coming from a relatively small number of subjects, there are a large number of skills and videos evaluated. In total, n=170 videos were rated, with the obvious limitation that not all those data points are not independent : -each participant contributed 5 goals at pre and again post – our analyses accounted for this – and we have now a discussion of this  to the limitations of the study.

Reviewer: The study population is heterogenous, for example BHC and CP are quite different, yet they are grouped together. The description of the limitations of the study is very brief.

Author’s response: Indeed, the population is relatively heterogeneous, but as we were interested in evaluating a rating scale that is of lesser importance here than it might be elsewhere. Further, the findings of the parent study, and indeed a number of CO-OP studies suggest that outcomes for CO-OP remain positive across diagnostic groupings. We have expanded on this in the limitations section. We agreed that the population is heterogeneous in nature.

Reviewer: The authors should comment whether the greater change for PQRS-I over PQRS-G meant that PQRS-I was more accurate?

Author’s response: thank you for this interesting comment. The term ‘accurate’ as applied to the changes noted is an interesting one. In response to reviewer 1, we have re-worked the discussion section clarifying that baseline means were significantly different for the two scales, even though the means at post-intervention were not significantly different. That the amount of change captured by the two scales was also significant was interesting, and a bit difficult to interpret, but likely due to the pre-test differences.  The findings suggest that the change captured by PQRS-I over PQRS-g indicates that PQRS-I was yields a larger change score which is meaningful to patients. As pointed out by reviewer 1 the score range for PQRS-I is wider than that of PQRS-G, we don’t consider this as more accurate – just perhaps more sensitive to change. Generally, in measurement, accuracy is related to standard error of measure – we did not evaluate that, so can’t comment on accuracy.

Reviewer: Further clinical details may be helpful for all patients. No clinical details are given for Child 1 (Table 1).

Author’s response: thank you. We will be willing to add further clinical details for all patients. What clinical information will the reviewer like to see further detailed? We have data on motor scores, dystonia severity, limbs affected, years lived with dystonia, whether MRI was normal or not; etc. Could we please have clarification about what further information would be useful?

Unfortunately, diagnosis for the group with aetiology ‘idiopathic’ cannot be given as this is a group of children for whom we do not have a cause of their dystonia.

Reviewer: It would be helpful for the authors go into more detail into what the intervention was.

Author’s response: we have expanded this in the introduction sections at the reviewer’s request

Reviewer: Could the authors discuss inter and intra-rater variability?

Author’s response: To the best of our knowledge inter and intra-rater variability is just another term for inter and intra-rater reliability – in our search for ‘variability’ studies we found the term very rarely used, and, when used, the methods in variability studies were the same as those used for reliability, the methods we used here[1].  We reported on inter-rater reliability; it was established for each of the scorer and systems with expert raters. We reported the ICCs in the manuscript. We did not evaluate intra-rater reliability.   We are curious to know what have we missed in our search of the psychometric properties literature with regard to inter and intra-rater variability? 

[1] Inter-rater variability was calculated by the intraclass correlation coefficient (ICC) and kappa (Fleiss) statistics.13   Journal List Tremor Other Hyperkinet Mov (N Y) v.7; 2017 PMC5618111  13. Kottner J, Audigé L, Brorson S, Donner A, Gajewski BJ, Hróbjartsson A, et al. Guidelines for reporting reliability and agreement studies (GRRAS) were proposed. J Clin Epidemiol. 2011;64:96–106. doi: 10.1016/j.jclinepi.2010.03.002. [PubMed] [Google Scholar]

Round 2

Reviewer 2 Report

The authors have addressed the major points raised by the review process.